# Morphological Characterization of Fossil *Vitis* L. Seeds from the Gelasian of Italy by Seed Image Analysis

**DOI:** 10.3390/plants13101417

**Published:** 2024-05-20

**Authors:** Mariano Ucchesu, Edoardo Martinetto, Marco Sarigu, Martino Orrù, Michela Bornancin, Gianluigi Bacchetta

**Affiliations:** 1Institute of Evolution Sciences of Montpellier (ISEM) UMR 5554, Université de Montpellier, CNRS, IRD, EPHE, Place Eugène Bataillon, 34090 Montpellier, France; marianoucchesu@gmail.com; 2Dipartimento di Scienze della Terra, Università di Torino, Via Valperga Caluso 35, 10125 Torino, Italy; edoardo.martinetto@unito.it (E.M.); michela.bornancin@edu.unito.it (M.B.); 3Centro Conservazione Biodiversità (CCB), Dipartimento di Scienze della Vita e dell’Ambiente (DISVA), Università degli Studi di Cagliari, Viale Sant’Ignazio da Laconi, 13, 09123 Cagliari, Italy; martino.orru@gmail.com (M.O.); bacchet@unica.it (G.B.)

**Keywords:** fossil *Vitis*, seed image analyses, Vitaceae, taxonomy

## Abstract

The discovery of well-preserved fossil *Vitis* L. seeds from the Gelasian stage in Italy has provided a unique opportunity to investigate the systematics of fossilized *Vitis* species. Through seed image analyses and elliptical Fourier transforms of fossil *Vitis* seeds from the sites Buronzo−Gifflenga and Castelletto Cervo II, we pointed out a strong relationship to the group of extant Eurasian *Vitis* species. However, classification analyses highlighted challenges in accurately assigning the fossil grape seeds to specific modern species. Morphological comparisons with modern *Vitis* species revealed striking similarities between the fossil seeds and *V. vinifera* subsp. *sylvestris*, as well as several other wild species from Asia. This close morphological resemblance suggests the existence of a population of *V. vinifera* sensu lato in Northen Italy during the Gelasian. These findings contributed to our understanding of the evolution and the complex interplay between ancient and modern *Vitis* species.

## 1. Introduction

The Vitaceae family consists of approximately 14 genera and 900 taxa distributed in different areas of the world, including Asia, Africa, Australia, and some Pacific islands, with a few genera located in temperate regions [1].

The genus *Vitis* L. comprises approximately 60 species, with at least 14 species and three named hybrid taxa native to North America and Caribbean region, one species complex in Europe (including the domesticated grape *V. vinifera* subsp. *vinifera*, and the wild *V. vinifera* subsp. *sylvestris*), and 37 species in China [2,3,4]. Phylogenetic analyses have confirmed the monophyletic nature of *Vitis*, consisting of two subgenera: subgenus *Muscadinia* Planch. (two species), found primarily in the southeastern United States, the West Indies, and Mexico, and subgenus *Vitis*, which encompasses the majority of species with a wide distribution across the Northern Hemisphere [2,3,4].

Phylogenetic studies conducted by Liu et al. [3] allowed us to reconstruct the origin of the genus *Vitis*. According to the results of this study, that used five plastid and two nuclear markers, it has been highlighted that the *Ampelocissus-Vitis* clade is composed of five main lineages, within which *Vitis* included two subgenera, each as a monophyletic group. In the same work, it is stated that *Vitis* originated in North America at approximately 39.4 Ma and subsequently migrated to Eurasia in the late Eocene (37.3 Ma), while, according to the study by Wan et al. [2], the divergence of the two subgenera *Euvitis* Planch. and *Muscadinia* Planch. took place approximately 18 Ma and was followed by the spreading and differentiation of the species, which became more evident during the tectonic and climatic changes of the Quaternary Period (starting from 1.3 Ma).

During the Pleistocene cold intervals, the populations of grapevine were isolated, producing consequential genetic diversification through allotropic speciation [5], as *Vitis vinifera* subsp. *sylvestris* (Gmelin) Hegi (hereafter, *V*. *vinifera sylvestris*) was found to be a sister to all Asian species and to be one of the many oldest living Eurasian species [6]. Even if seeds very similar to those of *V*. *vinifera sylvestris* are known before the major Pleistocene glaciations, it does not seem to be correct to apply the name of this subspecies to such fossils. In the case of sound evidence of a close relation to the living taxon, a possibility would be to name them *V. vinifera* sensu lato.

In interglacial periods, important redistributional phenomena occurred repeatedly, as did the isolation of plants in areas prone to adverse conditions (shelter), which has been fundamental to the evolution of the species [7,8]. The Caucasus, the Iberian Peninsula, Italy and Sardinia could have played a key role in protecting the genetic diversity of *V. vinifera sylvestris* and thus allowing the quick colonization of Central and Northern Europe during the postglacial period, when those places were subject to the range expansion of this taxon [9]. Climatic refugia [10] and vectors such as men, birds, foxes, bears, and turtles played key roles in the recolonization of a wide range of habitats and surfaces in the Mediterranean Basin and a limited number of places in Central Europe [9,11,12].

*Vitis* seeds exhibit specific morphological characteristics and small morphological modifications often reflect infrafamilial relationships that could be used to determine evolutionary and phytogeographic divergence [13]. The morphological characteristics of seeds in the genus *Vitis*, such as the dorsal chalaza and a pair of depressions called ventral infolds, have been systematically studied [13,14,15,16,17].

A particularly critical set of fossil seeds of *Vitis*, which occurs abundantly in the Neogene of Europe, was indicated as similar to the extant *V. vinifera sylvestris* [18,19,20]. Kirchheimer [21], based on morphological characteristics, identified as “*Vitis* cf. *silvestris* Gmelin” the fossil *Vitis* seeds from several European localities, e.g., the Upper Pliocene layers of Reuver in Limburg (The Netherlands), the Pliocene lignite-bearing deposits of the Wetterau (Hesse, Germany), and the Pliocene layers of Kroscienko (Neumarkt, Poland). Later, the same author [18] proposed, in a much more unclear treatment without illustrations, a new fossil-species name, *V. parasylvestris* Kirchh., for fossil seeds similar to the extant *V. vinifera sylvestris*. In a separate popular publication [22], the same author figured two fossil seeds only from the ventral side but failed to indicate the name *V. parasylvestris*. However, in his comprehensive book [19], he considered that these two seeds were the types and that the species name was validly published and should be applied to all of the Neogene records of seeds similar to those of *V. vinifera sylvestris*. Such an approach was later criticized by Mai and Walther [23] and Geissert et al. [20]. These authors accepted the fossil-species *V. parasylvestris* but pointed out that this name should only be applied to a limited number of Neogene fossils, those in which “the chalaza and the dorsal side are very different [in comparison to *V. vinifera sylvestris*], but also the prolongation of the basis and the ventral invaginations are otherwise” [23]. Fossil seeds which referred to the extant taxon “*V. sylvestris”* occur in the Upper Miocene, according to Mai and Walther [23], only in Eastern Europe. Conversely, in the Pliocene and Early Pleistocene, they are widely distributed in all European territory and occur above the present northern limit of *V. vinifera sylvestris* in the Holsteinian and Eemian intergacials.

Overall, the differential characters of *V. parasylvestris* and “*V. sylvestris”* were never described with an acceptable precision, and, in those fossil assemblages that provided hundreds of seeds, the two types are mixed and linked by intermediate forms (e.g., in the Mio-Pliocene “Saugbaggerflora”) [20]. Indeed, further studies on specimen-rich fossil assemblages of seeds similar to those of *V. vinifera sylvestris* would be needed.

Recently, different studies conducted by the seed image analysis technique confirmed the importance of identifying both modern and archaeological grape seeds [24,25,26,27,28,29,30,31,32,33,34,35,36] and we deemed it useful to carry out a seed image analysis on non-archaeological fossil assemblages that consisted approximately of one hundred specimens. In the Italian fossil seed collections, this requirement was only fulfilled by Early Pleistocene seed assemblages from the Cervo River section, in NW Italy. Approximately 1 km south of Castelletto Cervo, in the province of Biella (Italy), the erosion caused by the Cervo River exposed sandy–silty sediments of palaeobotanical interest [37,38]. Within this stratigraphic succession, 242 seeds of the genus *Vitis* were recovered.

Using a seed image analysis, the aims of this study were to characterize fossil *Vitis* seeds and to explore the morphological relationships between fossil seeds and modern wild species of the genus *Vitis* originating from North America, Europe, and Asia. The objectives of the study also extend to questions about the evolutionary and phytogeographic divergences of the genus *Vitis*.

## 2. Palaeontological Background of the Study Area

The studied assemblages of fossil *Vitis* seeds were recovered from two plant-bearing beds of the Cervo River section (Figure 1), cropping out near the villages of Castelletto Cervo and Gifflenga, even if the second locality was named [39] after the largest neighbouring village of Buronzo, and will be referred to here as Buronzo-Gifflenga. In both sites, the sediments are mainly composed of paralic sands, muds, and brown coals with mummified plant remains. The animal fossils are represented only by casts of the autochthonous claim *Cerastoderma* Poli sp., which is abundant in a single layer and probably lived in a brackish environment. Plant macrofossils include trunks, small stumps, fern rhizomes, and leaf compressions. Mummified fruits and seeds are abundant in a few layers, and the largest specimens can be easily collected on the outcrop surface: fruits of *Juglans bergomensis* (Bals.-Criv.) A. Massal., and seeds of *Euryale* Salisb. sp., *Magnolia cor* R.Ludw., and *Quercus* L. sp., documented by cupules. A particular layer with *Trapa* L. sp. fruits is so rich in such remains that it has been named the “*Trapa* layer” [40]. In the carpological assemblages gathered from the two *Vitis*-bearing layers («Castelletto Cervo II flora» and «Buronzo-Gifflenga flora»: Figure 1), the “HUTEA” elements [41], which characterize the Pliocene assemblages in northern Italy, are completely lacking. However, several species with exotic affinities are still present: *Euryale nodulosa* C.Reid & E.Reid*, Liriodendron geminatum* Kirchheim., *Magnolia cor, Juglans bergomensis, Phellodendron elegans* C.Reid & E.Reid, and *Symplocos* cf. *paucicostata* (C.Reid & E.Reid) Mai & Martinetto.

The Cervo River section, among the Italian stratigraphic sections, is one of the richest in carpological fossil remains and one of the few that provides several stratigraphically superposed assemblages. These have been the object of either preliminary taxonomic analyses [39,40,42,43] or more accurate systematic treatments [37,38]. The lower part of this ca. 220 m-thick succession is dated to the Zanclean based on marine palaeontological records [44], whereas its upper part has no dating elements, apart from the palaeoflora data. The sediments are not suitable for a continuous pollen record because of the long intervals made up of oxidized gravel and sand deposits; thus, only short fine-grained portions of the upper part of the succession have been analyzed by palynologists [45]. The age of the succession, based only on palaeobotanical records, suggested a long-term coverage [39], at least late Zanclean–Gelasian (ca. 4 to 2 Ma), if not late Zanclean–Calabrian (ca. 4 to 1 Ma).

The abundant plant macrofossil assemblages studied thus far in the Cervo River section are widely scattered from top to bottom and plotted within different climatic phases [39]. The floral character of the five lower assemblages is rather homogeneous and shows a typical Zanclean–early Piacenzian composition [46]. Floras from overlaying layers of the Cervo River section (Terzoglio III-Castelletto Cervo I) point to a Piacenzian age according to the biochronological criteria described in Martinetto et al. [38,46]. The upper 50 metres of the Cervo River succession are separated by older layers by a large fault, which introduces a wide, but not precisely quantifiable, gap in the stratigraphic succession. The fault separates two blocks of sediments rich in plant fossils, which are considerably different in the upthrown block (Castelletto Cervo I flora) versus the downthrown block (Castelletto Cervo II and Buronzo−Gifflenga floras).

Plant assemblages of the downthrown block contain some relevant taxa that may be considered as biochronological indicators. In particular, the occurrence of *Actinidia*, *Azolla tegeliensis*, *Menispermum*, and *Pseudolarix* in the Castelletto Cervo II flora would suggest a Gelasian rather than Calabrian age, and the late Gelasian age (ca. 1.9 Ma) of the younger Buronzo−Gifflenga flora is suggested by the co-occurrence of *Azolla tegeliensis* and *Azolla filiculoides* [47]. A thick red palaeosol between the two layers can be associated with a consistent time gap, so that the Buronzo−Gifflenga flora should be at least a few hundred thousand years younger than the Castelletto Cervo II flora (within the time interval of 2.5–1.9 Ma).

## 3. Results

### 3.1. Comparison of Fossil Vitis Seeds

To compare the fossil assemblages of the Buronzo–Gifflenga flora and the Castelletto Cervo II flora, a PCA was performed using 24 coefficients (Figure 2). The PC1–PC2 (41.4% of the total variance) biplot showed no clear difference between the two fossil grape seed assemblages: Both were distributed in the centre of the plot (Figure 2).

### 3.2. Comparison of Fossil Seeds with the Modern Vitis Species

Based on the previous PCA results, both fossil grape seeds were considered a single group and were compared by LDA to two groups of modern *Vitis* species; one included Eurasian species, while the other included all North American accessions (Appendix A).

In the LDA analysis, the fossil grape seeds, considered as an unknown group, were assigned to the group of Eurasian *Vitis* species, for which 94.3% of the species were correctly classified and only 5.7% were attributed to North American *Vitis* species (Table 1, Figure 3).

To determine which modern Eurasian *Vitis* species exhibited a close relationship with the fossil grape seeds, an additional LDA analysis was conducted. The fossil grape seeds were compared with the modern *Vitis* species from Eurasia considered as individual accessions. The LDA results showed a high similarity to those of *V. vinifera sylvestris* (41.4%), while the remaining fossil grape seeds were assigned to *V. amurensis* (11.4%), *V*. *heyneana* (11.4%), *V. ficifolia* (20.0%), and *V. romanetii* (14.3%) (Table 2, Figure 4).

Finally, to achieve a more accurate classification of fossil grape seeds, a further LDA was conducted, focusing only on the Eurasian species exhibiting a close morphometric correlation with the fossil grape seeds identified in the previous LDA (Table 2). In this case, the LDA revealed that the fossil grape seeds were correctly classified as *V. ficifolia* with a percentage of 32.9% and *V. vinifera sylvestris* with a percentage of 32.9%. Furthermore, a portion of the fossil grape seeds was classified as *V. amurensis* (22.9%), with only a minor portion assigned to *V. heyneana* and *V. romanetii* (5.7%) (Table 3, Figure 5).

Subsequently, considering a probability threshold of *p* ≥ 0.90 for the distribution of fossil seeds in the previous LDA classification, the analysis showed that 37.1% of the fossil seeds were not allocated to any of the five species. However, 25.7% of the fossil seeds were assigned to *V. vinifera sylvestris*, and 15.7% and 14.3% to *V. ficifolia* and *V. amurensis*, respectively. Only 5.7% and 1.4% of the fossil seeds were assigned to *V. heyneana* and *V. romanetii*, respectively (Figure 6).

Based on the previous results obtained from the discriminant analysis where grape fossils were mostly assigned to *V. vinifera sylvestris*, we conducted a further discriminant analysis by comparing the fossil grape seeds with two groups of *V. vinifera sylvestris* populations from the Western and Eastern ecotype. The first group belonged to the ecotype present in Western Europe (Italy, France, and Spain), while the second group belonged to the ecotype present in Western Asia (Georgia). The LDA results showed a high similarity to the Western ecotype (78.6%), while the remaining fossil grape seeds (21.4%) were assigned to Eastern ecotype (Table 4, Figure 7).

## 4. Discussion

Fossil seeds belonging to the Vitaceae family are well-represented throughout the European Tertiary, suggesting that there was a widespread distribution of the mother plants [13,19]. Additionally, the occurrence of many separate taxa (e.g., three genera and six fossil-species reported by Czaja [48], particularly in the Miocene [19] suggests a consistent diversification during the Palaeogene. Molecular dating places the origin of *Vitis* in the Palaeogene, suggesting that the common ancestor of *Vitis* originated in North America [2]. This hypothesis seems to be supported by the findings of fossil *Vitis* seeds in the Eocene deposits of Northwestern America, while no evidence comes from Southeast Asian before the Pliocene [2,49]. However, sound evidence for the presence of *Vitis* in Europe is also provided by Eocene fossil seeds of *V. messelensis* [50].

The patterns of morphological variation exhibited by seeds of the Vitaceae family may be correlated with intrafamilial relationships; thus, fossil *Vitis* seeds can be useful in addressing questions about evolutionary and phytogeographic divergences [2].

Since Vitaceae seeds are characterized the presence of both the paired ventral infolds and the dorsal chalaza that are not found in seeds of other families, their identification is quite reliable. While the family-level identification is facilitated by the characteristics just described, assigning them to specific genera or species is rather challenging. Moreover, additional limitations for the exact identification of fossil seeds are due to the limited availability of materials and the poor preservation of fossil remains, which do not retain all seed characteristics [13].

Recent studies, that have employed a morphometric seed image analysis, proved to be effective in distinguishing wild grapes from domestic ones in both modern and archaeological seeds [31,33,34,35,36,51,52]. The same methodology was extended beyond mere species identification, aiming to differentiate groups of domestic grape varieties or specific wild grape populations [24,25,26,28,30,31,33,34,35,36,53,54,55,56].

In this study, we applied a seed image analysis and elliptical Fourier transforms to non-archaeological fossil *Vitis* seeds, comparing them with a comprehensive database that included species of *Vitis* from North American and Eurasian origins.

The discovery of well-preserved fossil *Vitis* seeds in the Buronzo−Gifflenga and Castelletto Cervo II floras of NW Italy, of probable Gelasian age, allowed us to explore, for the first time, the systematics of non-archaeological fossils.

The morphometric comparison of both accessions of fossil grape seeds found in the Buronzo−Gifflenga and Castelletto Cervo II floras revealed that they belong to the same *Vitis* species. These materials came from two different stratigraphic layers: the seeds from Castelletto Cervo II were found at the base of the succession assigned to the lower part of the Gelasian stage, dating approximately around 2.5–2.2 million years ago, and those from Buronzo−Gifflenga were found in the upper Gelasian layers, with an approximate date of about 2.1–1.9 million years ago. Our analyses showed that the morphology of these fossil seeds remained unchanged for a relevant time, possibly approaching half a million years.

The comparative analysis between the Italian fossil *Vitis* seeds and the modern materials that had been grouped as Eurasian and North American accessions allowed us to classify the fossil *Vitis* seeds in the Eurasian species groups. Indeed, according to the results obtained by Wan et al. [2], the divergence between Eurasian and American wild grape species would have occurred approximately 11.2 million years ago, and our findings are in agreement with such a hypothesis, showing that the morphology of the Italian fossil *Vitis* seeds does not match with the wild *Vitis* species native to North America. Moreover, a recent genetic analysis established the biogeographic disjunctions of the subgenus *Vitis* between North America and Eurasia [3].

A further comparative analysis of individual accessions of 11 wild grape species from Eurasia showed that the morphology of fossil *Vitis* seeds was highly similar to modern *V*. *vinifera sylvestris* and four other wild species from Asia (*V. ficifolia*, *V*. *amurensis*, *V. heyneana*, and *V. romanetii*). However, when compared with the fossil grape seeds of only these five species, the fossil *Vitis* seeds appear to be morphologically very similar to *V. vinifera sylvestris*, and to two Asian wild grape species (*V. amurensis* and *V. ficifolia*). Upon further analysis, where we considered a classification probability threshold of *p* ≥ 0.90, LDA showed that a high percentage of the fossil *Vitis* seeds were assigned to the *V. vinifera sylvestris*. The same analysis also showed that a high percentage of specimens was not assigned to any of the modern *Vitis* species. This could be related to the distortion in the morphology of the fossil seeds, which prevented an accurate classification of the samples.

However, our study confirms the presence of the genus *Vitis* with morphological characteristics attributable to the extant *V. vinifera sylvestris* in the Italian Gelasian deposit. Based on the results obtained from our study, it is plausible to consider that the fossil *Vitis* seeds found in the Gelasian layers of northern Italy may have belonged to the lineage of the present Eurasian wild grapevine and, in particular, with the current populations’ ecotype present in Western Europe.

However, we cannot assert with absolute certainty that they belonged to *Vitis vinifera* L. subsp. *sylvestris*, since this subspecies could be the result of a bottleneck effect due to various population geographic isolation processes that occurred over the last 400,000 years [57]. The study of Zecca et al. [6] demonstrated that the diversification of wild grapes is a continuous and complex process that has involved both the Neogene and the Quaternary periods, encompassing both geographical and climatic forces.

Therefore, we deem it appropriate to classify these fossil seeds as *Vitis vinifera* L. sensu lato. The above-cited identification of various fossil grape seeds apparently similar to those of modern European wild grape, in the European Pliocene (see Introduction), suggests that our Gelasian specimens may not represent the oldest fossil record of *V. vinifera* sensu lato in Europe.

## 5. Materials and Methods

### 5.1. Modern Grape Seed Accessions

A total of 536 modern grape seeds were obtained from 17 *Vitis* species; 11 taxa from Asia and Europe (hereafter Eurasian) and 6 species from North America were used as reference materials for comparison with fossil seeds (Appendix A, Figure 8). Modern materials were obtained from the collection of Institut des Sciences de l’Évolution de Montpellier (ISEM), University of Montpellier, (France), Sardinian Germplasm Bank (BG-SAR) of the University of Cagliari (Italy), Arnold Arboretum of Harvard University of Boston (Massachusetts, USA), Julius Kühn Institute (JKI) of Quedlinburg (Germany), and from Nat’l Clonal Germplasm Rep—Tree Fruit & Nut Crops & Grapes (NCGR) of Davis (California, IL, USA).

### 5.2. Fossil Grape Seed Accessions

The fossil seeds analyzed in this study display central chalaza positioned on the dorsal surface, a clearly visible chalaza-apex groove, and short linear ventral infolds, all of which closely align with the characteristics of the genus *Vitis* [13,14].

The fossil diaspores of *Vitis* consisted of 202 seeds found in the section that yielded the Buronzo−Gifflenga flora and 40 seeds found in the section that yielded the Castelletto Cervo II flora (Figure 1). Each set of seeds was extracted by a single sediment sample collected from a definite layer. These samples were processed with 5% hydrogen peroxide to improve sediment disaggregation and determine a partial floatation of the fossil seeds. After the reaction was complete, the floating fraction was sieved separately from the sunken material (for a final mesh size of 0.3 mm). Finally, fruits, seeds, and related organs were picked from the residue of both fractions. The sediment volume of the samples analyzed approached 20 dm^3^.

To minimize classification errors, we selected fossil specimens that showed completely intact morphology (presence of beak, and absence of deformation or breakage). In total, 70 well-preserved fossil *Vitis* seeds were utilized (Figure 9).

### 5.3. Morphometric Analysis

Digital images of dorsal views of modern and fossil grape seeds were acquired using a flatbed scanner (Epson Perfection V600 photo, Suwa, Japan), with a digital resolution of 600 dpi for a scanning area not exceeding 1024 × 1024 pixels [58]. All the images were converted to black silhouettes using the software package ImageJ v. 1.54 (http://rsb.info.nih.gov/ij (accessed on 30 March 2024), and outline analyses were performed using elliptical Fourier transforms (EFTs) following the method described in Terral et al. [53]. This method allows us to describe the geometry of the seed boundary and converts the outline of an object into shape descriptors (Fourier coefficients = EFDs) [59]. The EFT transforms (x; y) co-ordinates of the outline into “Fourier coefficients”, which are then treated as multivariate variables. Initially, 360 equidistant points were sampled along the curvilinear abscissa. Subsequently, the outlines underwent normalization for size, rotation, position, and the first point. Consistent with previous studies [60], the first six harmonics were employed to describe each view, resulting in 24 coefficients (four coefficients per harmonic, for one view) useful for discriminating between *Vitis* species. This decision, based on six harmonics, strikes a balance between accurately describing shape (capturing more than 95% of the total harmonic power) and minimizing measurement errors, which tend to increase with harmonic rank [60]. The outline analyses were conducted using ImageJ.

### 5.4. Statistical Analysis

To investigate the overall morphological variation of fossil seeds under study, principal component analysis (PCA) was first applied. After, focusing on uncorrelated PCA scores, stepwise linear discriminant analysis (LDA) was applied using leave-one-out cross-validation. This approach is commonly used to classify/identify unknown groups characterized by quantitative and qualitative variables [61], finding the combination of predictor variables with the aim of minimizing the within-class distance and maximizing the between-class distance simultaneously, thus achieving maximum class discrimination [62]. Statistical analysis was performed using IBM SPSS software v. 20.0 (Statistical Package for Social Science) (SPSS, Inc., for Windows, Chicago, IL, USA).

## 6. Conclusions

This research represents the first study in which seed image analyses and elliptical Fourier transforms were applied for the characterization of fossil *Vitis* seeds.

Morphological comparisons with modern *Vitis* species unveiled remarkable resemblances between the fossil seeds and *V. vinifera* subsp. *sylvestris*, alongside several other wild species from Asia. This close morphological similarity implies the presence of a population of *V. vinifera* sensu lato in Northern Italy during the Gelasian period. In this work, we have demonstrated the potential of seed image analysis to be successfully applied to fossil plant materials and proved how this methodology has been able to clarify information regarding the identification of fossil *Vitis* seeds.

The validity of this methodology is closely related to two important factors that must be taken into consideration when we want to characterize fossil materials: The first is the quantity of the materials; since the results are obtained on a statistical basis, it is essential to be in possession of a sufficiently large number of both fossil and modern materials. The second aspect is the quality of the material, which must not show excessive morphological deformations that could produce inaccurate parameters that could affect the results.

We hope that this work may stimulate further research in this field and contribute to understanding the evolutionary and phytogeographic divergences for *Vitis* and other fossil taxa.

## Figures and Tables

**Figure 1 plants-13-01417-f001:**
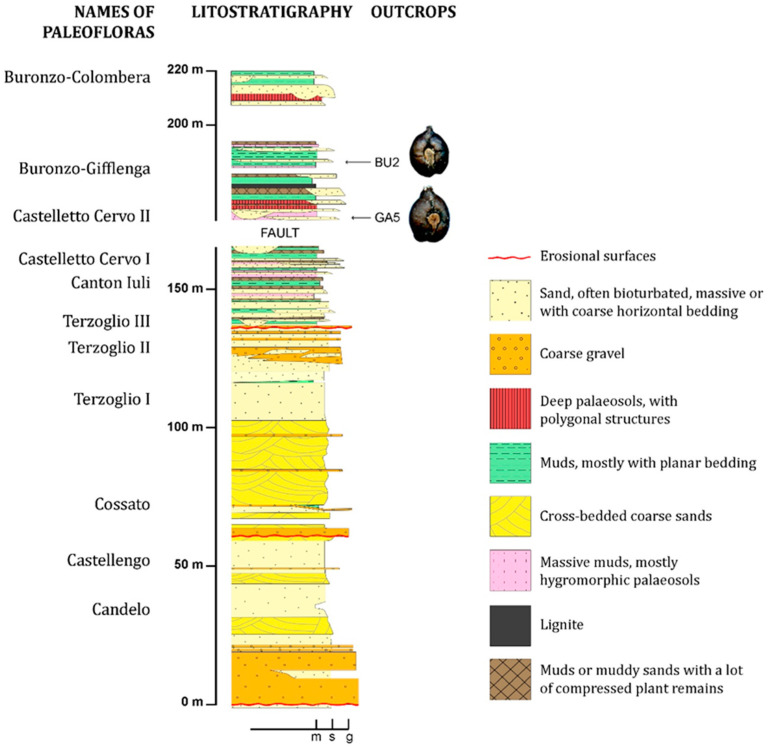
Composite stratigraphic log of the sedimentary succession cropping out along the Cervo River. The strata gently slope toward southeast, and, therefore, the oldest deposits (Zanclean) are located in the northwestern part of the valley (Candelo−Cossato), whereas the youngest (Gelasian and/or Calabrian) to the southeast (Castelletto Cervo II and Buronzo−Gifflenga, incl. *Vitis* layers). Modified from Martinetto and Festa [40]. m, muds; s, sands; g, gravels; GA5 and BU2: labels of the two fossiliferous beds with *Vitis* seed assemblages.

**Figure 2 plants-13-01417-f002:**
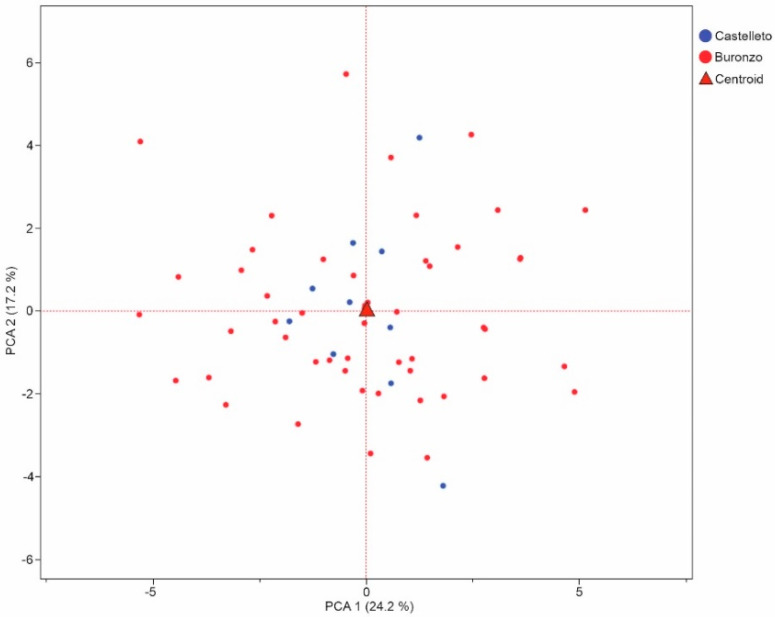
Results of principal component analysis using the 24 coefficients of 60 fossil grape seeds from the Buronzo–Gifflenga flora and 10 fossil *Vitis* seeds from Castelletto Cervo II flora.

**Figure 3 plants-13-01417-f003:**
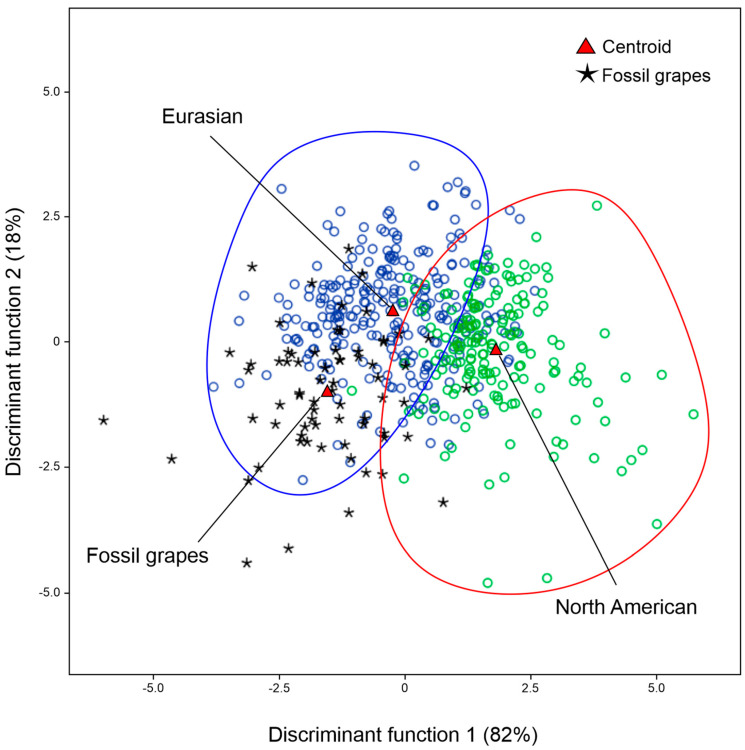
LDA graphical representation of the allocation of fossil grape seeds considered an unknown group among the two groups of *Vitis* species.

**Figure 4 plants-13-01417-f004:**
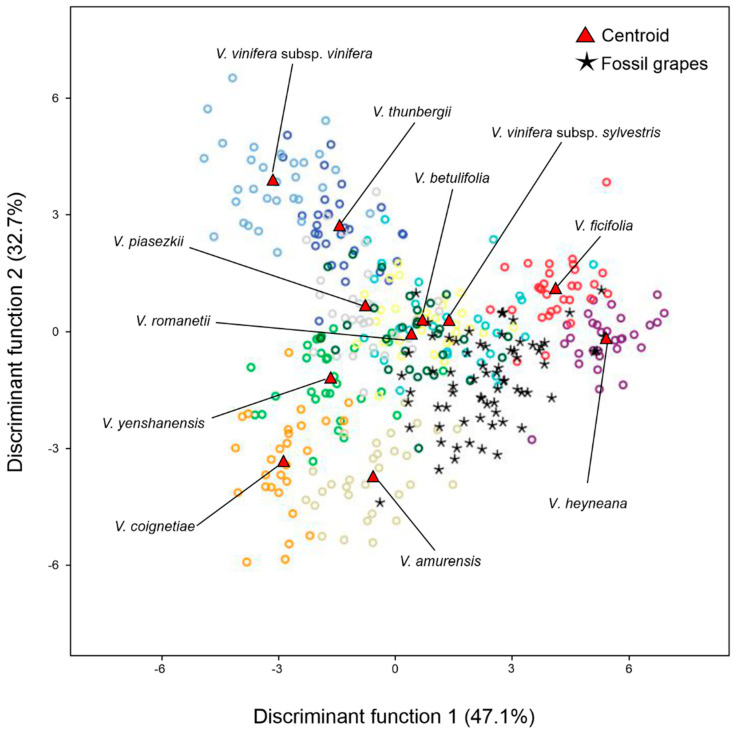
LDA graphical representation of the allocation of fossil grape seeds considered to be an unknown group in the 11 accessions of Eurasian *Vitis* species.

**Figure 5 plants-13-01417-f005:**
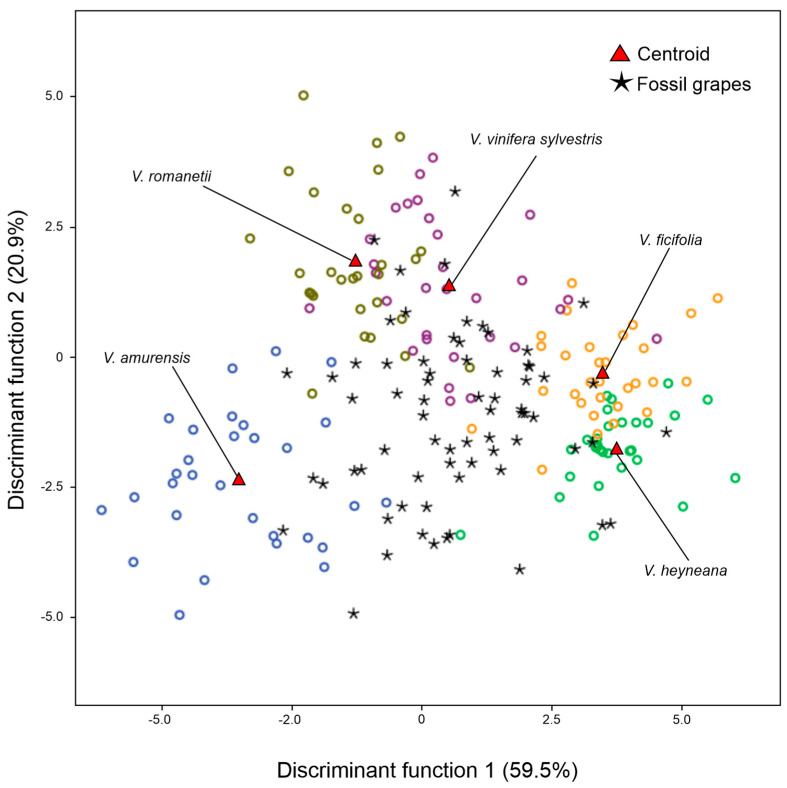
LDA graphical representation of the allocation of fossil grape seeds considered unknown group in the five accessions of Eurasian *Vitis* species.

**Figure 6 plants-13-01417-f006:**
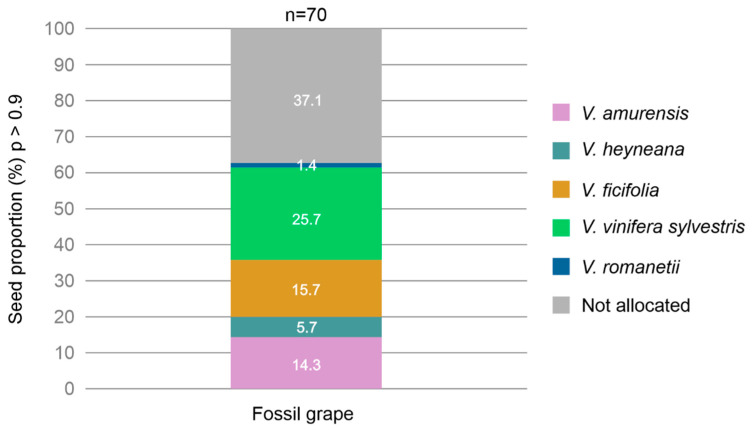
LDA percentage distribution (*p* ≥ 0.9) of fossil grape seeds according to the allocation to the five Eurasian *Vitis* species.

**Figure 7 plants-13-01417-f007:**
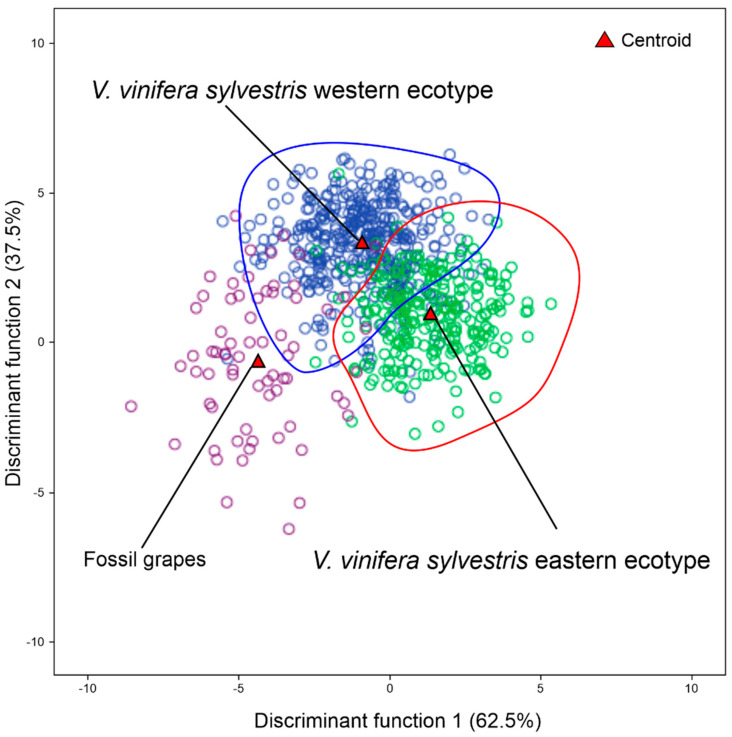
LDA graphical representation of the allocation of fossil grape seeds considered an unknown group among the two ecotype groups of *V. vinifera sylvestris*. Western ecotype: Spain, France, and Italy; Eastern ecotype: Georgia.

**Figure 8 plants-13-01417-f008:**
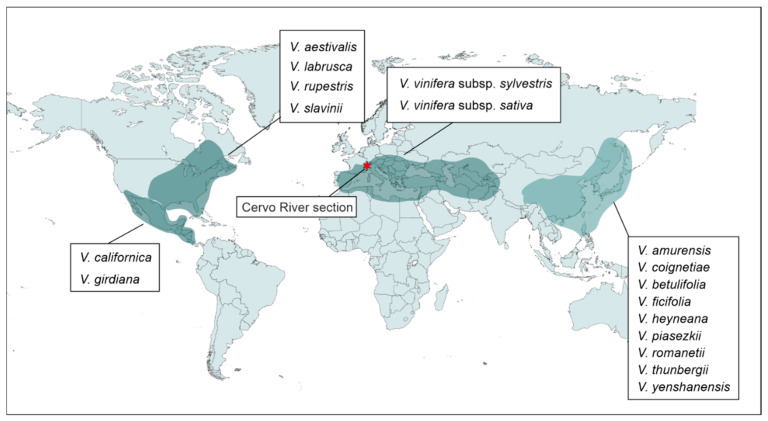
Geographic distribution of *Vitis* species used in this study and location of the Cervo River section.

**Figure 9 plants-13-01417-f009:**
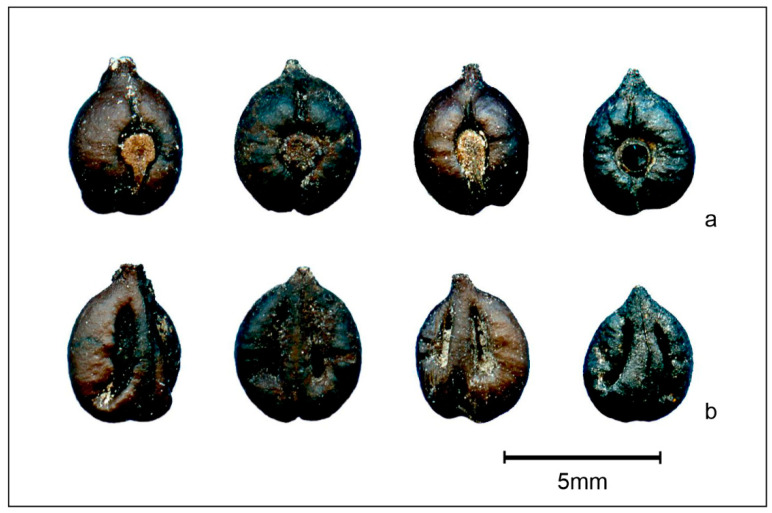
Representative image of some fossil *Vitis* seeds from Buronzo-Gifflenga flora used in this work ((**a**) = dorsal; and (**b**) = ventral).

**Table 1 plants-13-01417-t001:** Correct classification percentage between the fossil seeds, considered the unknown group, and the two *Vitis* species group.

	Eurasian *Vitis*	North American *Vitis*	Total
Eurasian *Vitis*	77.7	22.3	100
North American *Vitis*	12.2	87.8	100
Fossil grapes	94.3	5.7	100

79.7% of cross-validated grouped cases correctly classified.

**Table 2 plants-13-01417-t002:** Correct classification percentage between the fossil seeds, considered an unknown group, and the 11 *Vitis* species from Eurasia.

	*V. thunbergii*	*V. yenshanensis*	*V. amurensis*	*V. heyneana*	*V. betulifolia*	*V. ficifolia*	*V. vinifera sylvestris*	*V. piasezkii*	*V. vinifera*	*V. romanetii*	*V. coignetiae*	Total
*V. thunbergii*	80	3.3	-	-	-	-	-	10	6.7	-		100
*V. yenshanensis*	-	83.3	3.3	-	-	-	-	10	-	-	3.3	100
*V. amurensis*	-	-	90	-	3.3	-	-	-	-	-	6.7	100
*V. heyneana*	-	-	-	100	-	-	-	-	-	-		100
*V. betulifolia*	-	-	-	-	96.7	-		-	-	3.3		100
*V. ficifolia*	-	-	-	6.7	-	93.3	-	-	-	-		100
*V. vinifera sylvestris*	6.7	-	-	-	3.3	10	80	-	-	-		100
*V. piasezkii*	10	3.3	3.3	-	-	-	-	76.6	3.3	3.3		100
*V. vinifera*	10	-	-	-	-	-	-	3.3	86.7	-		100
*V. romanetii*	-	3.4	-	-	6.9	3.4	-	-	3.4	82.8		100
*V. coignetiae*	-	7.1	3.6	-	-	-	-	-	-	-	89.3	100
Fossil grape	-	-	11.4	11.4	1.4	20	41.4	-	-	14.3	-	100

81.3% of cross-validated grouped cases correctly classified.

**Table 3 plants-13-01417-t003:** Correct classification percentages between the fossil seeds, considered an unknown group, and the five *Vitis* species from Eurasia.

	*V. amurensis*	*V. heyneana*	*V. ficifolia*	*V. vinifera sylvestris*	*V. romanetii*	Total
*V. amurensis*	93.3	-	-	-	6.7	100
*V. heyneana*	-	93.3	6.7	-	-	100
*V. ficifolia*	-	-	100	-	-	100
*V. vinifera sylvestris*	-	3.3	6.7	90	-	100
*V. romanetii*	3.4	-	-	3.4	93.1	100
Fossil grapes	22.9	5.7	32.9	32.9	5.7	100

91.3% of cross-validated grouped cases correctly classified.

**Table 4 plants-13-01417-t004:** Correct classification percentage between the fossil seeds, considered the unknown group, and the two ecotype groups of *V. vinifera sylvestris*.

	Western *V. vinifera sylvestris*	Eastern *V. vinifera sylvestris*	Total
Western *V. vinifera sylvestris*	81.1	18.9	100
Eastern *V. vinifera sylvestris*	18.9	81.1	100
Fossil grapes	78.6	21.4	100

81.0% of cross-validated grouped cases correctly classified.

## Data Availability

Data are contained within the article and Appendix A.

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
