# Peer review of "Morphological Characterization of Fossil Vitis L. Seeds from the Gelasian of Italy by Seed Image Analysis"

_plants, 2024, doi:10.3390/plants13101417_

Round 1
Reviewer 1 Report
Comments and Suggestions for Authors
Dear Authors,
The submitted manuscript titled „Morphological characterization of fossil Vitis seeds from the Gelasian of Italy by seed image analysis” contains very intersting results, which might interest an international audience. Nevertheless, I have found some imperfections, which-in my opinion-should be corrected or at least clarified before an eventual publication. I have listed them below:
1. In my opinion the choice of genus Vitis L. for investigations should be better justified.
2. Lines 32-34; I suggests to extend the description of genus Vitis L, especially refering to studied species.
3. Please check the order of references in the text. In lines 35 and 41 two different literature sources have number [2]., while numbers [3] and [4] are missing.
Author Response
Dear reviewer, thank you for your comments.
We attach the file with the answers.
You will find the changed parts in the manuscript in red color.

Reviewer 2 Report
Comments and Suggestions for Authors
In this manuscript the authors analyze morphology of fossil seeds of Vitis from the Gelasian stage of Italy using seed image analyses and Ellpitical Fourier Transforms. Their results show striking similarities between the fossil seeds and V. vinifera subsp. sylvestris. The research contributes to knowledge of evolution and biogeographic history of the Vitis genus. The references are adequate and the illustrations are clear and necessary. Overall it is a carefully prepared and generally well written paper.
Two suggestions:
1. I suggest the author explain more on the method of morphometric analysis.
2. I would be helpful to discuss the biogeographic history of Vitis a little bit.
Author Response

(The authors gave the same response as above.)
